

# Barriers to seeking consultation at public service and medical facilities among pregnant and postpartum women in Japan: a cross-sectional study

Sumiyo Okawa[1,2], Kaori Seino[1], Hiroyasu Iso[2] and Takahiro Tabuchi[3,4]

[1] Bureau of International Health Cooperation, National Center for Global Health and Medicine, Shinjuku-ku, Tokyo, Japan
[2] Institute for Global Health Policy Research, Bureau of International Health Cooperation, National Center for Global Health and Medicine, Shinjuku-ku, Tokyo, Japan
[3] Division of Epidemiology, School of Public Health, Tohoku University Graduate School of Medicine, Sendai, Miyagi, Japan
[4] Cancer Control Center, Osaka International Cancer Institute, Chuo-ku, Osaka, Japan

Corresponding author
Sumiyo Okawa, okawa.s@jihs.go.jp

## ABSTRACT

**Background**. Pregnant and postpartum women often experience parenting problems which may affect their mental health and children's health and development. However, their ability to seek consultation at public service or medical facilities remains unclear. This study aimed to identify the characteristics associated with women who refrain from visiting these facilities for consultation and their reasons for it.

**Methods**. This cross-sectional internet-based survey was conducted in Japan between July and August 2021, involving 7,326 women (1,639 pregnant and 5,687 postpartum women). The study outcome was defined as refraining from seeking consultation on family or parenting issues at public service or medical facilities despite a perceived need. We identified women's characteristics associated with refraining from consultation using multivariable logistic regression and conducted a descriptive analysis of 13 listed reasons for refraining from the consultation.

**Results**. The percentage of women who reported refraining from seeking consultation at public service or medical facilities was 8.6% and 5.1%, respectively. Common factors associated with refraining from seeking consultation at these facilities included having a child(ren) and being in the postpartum period, low health literacy, lack of partner support, and current disability. "Difficulty taking child(ren) to the consultation" and "uncertainty about the seriousness of the problem" were major reasons for refraining from consultation.

**Discussion**. Healthcare workers at public service and medical facilities should offer prenatal education on the importance of seeking help. An online consultation service and an improved facility environment may mitigate women's perceived barriers to seeking consultation.

## INTRODUCTION

Universal health coverage, defined as "all people have access to the full range of quality health services they need, when and where they need them, without financial hardship (*World Health Organization, 2024*)," is key to achieving Sustainable Development Goals 3: "Ensure healthy lives and promote well-being for all at all ages" (*United Nations, 2015*). Women and children are particularly vulnerable regarding access and utilization of essential healthcare. Approximately 47,000 mothers and 423,000 newborns in low- and middle-income countries died due to a lack of healthcare utilization (*Kruk et al., 2018*). Although most maternal and newborn deaths occur within the first month after childbirth in these countries (*Lawn et al., 2014*; *Say et al., 2014*), depression and anxiety are common morbidities during pregnancy and the postpartum period globally (*National Institute for Health and Care Excellence, 2014*; *Vogel et al., 2024*), which can impede mother-to-child interactions, maternal care, and child development (*Slomian et al., 2019*). Therefore, ensuring women's access to healthcare services for family, parenting, and child health issues is important from pregnancy through the early and late postpartum periods.

Poor access to healthcare services for women is not always caused by healthcare system failures but often stems from women's low capability or motivation to seek care. A review study reported major reasons for not seeking health care services: women tend to accept problems as part of normal motherhood, fear negative judgment, and face societal norms that often normalize maternal health issues (*Rouhi et al., 2019*). Studies conducted in low-, middle-, and high-income countries have reported various barriers and disparities in women's access to healthcare services, including age, ethnicity, healthcare costs, recognition of danger signs, transportation arrangement, and accessibility to a health facility (*Lassi et al., 2019*; *Mi et al., 2022*; *Wilcox, Levi & Garrett, 2016*). While the concept of access to healthcare services remains complex, a previous study proposed a framework with five dimensions (approachability, acceptability, availability and accommodation, affordability, and appropriateness) and five abilities that enable a person to access healthcare (the ability to perceive, seek, reach, pay, and engage) (*Levesque, Harris & Russell, 2013*).

Japan has an established healthcare system offering public health and medical services to women and their children at low costs. Women receive maternity care from pregnancy through delivery and up to 1 month postpartum at medical facilities, with expenses subsidized by the local government. Children receive home-visit health checks and regular checkups at pediatric clinics or public health facilities until 3 years of age (*Ministry of Health Labour and Welfare, 1998*). They also receive medical services under health insurance, supplemented by local government subsidies (*Children and Families Agency Government of Japan, 2023*) and various childcare support services provided by the local government (*Children and Families Agency Government of Japan, 2022*). While maternal and infant mortality rates are low, maternal mental health remains a critical public health issue. For example, a meta-analysis reported the period prevalence of prenatal depression at 14.0% in the second trimester and 16.3% in the third trimester. The period prevalence of postpartum depression was 15.1% in the first month, 11.6% at 1–3 months, 11.5% at 3–6 months, and 11.5% at 6–12 months after birth (*Tokumitsu et al., 2020*). During the COVID-19 outbreak,
maternal depression was observed in 30.3% of pregnant women and 24.2% of postpartum women (*Chen et al., 2023*), with mother-to-infant bonding difficulties at 26.1% (*Chen et al., 2024*). This suggests that women may encounter family and parenting issues, leading to subsequent mental health problems. However, access to public services or medical facilities for consultation has not been thoroughly investigated.

Therefore, this study aimed to (1) identify the characteristics of women associated with refraining from seeking consultation at public service or medical facilities and (2) examine reasons why women refrain from seeking consultation at these facilities.

## MATERIALS & METHODS

### Study design and participants

A nationwide cross-sectional internet-based survey was conducted as part of the Japan COVID-19 and Society Internet Survey (JACSIS). Study participants were selected from a pooled panel of an internet research agency (Rakuten Insight, Inc.) with approximately 2.2 million registered panelists (as of September 2022) (*Rakuten Insight Inc, 2022*). The participants were pregnant women expected to deliver by December 2021 and postpartum women with live singleton deliveries between January 2019 and August 2021. First, we conducted a screening survey and identified 14,086 women (2,425 pregnant and 11,661 postpartum women) who met our participant criteria. After sending invitation emails to these eligible women, 8,047 women (1,791 pregnant and 6,256 postpartum) participated in the survey between July and August 2021. We considered this the maximum number of voluntary participants we could obtain. Among these women, 7,326 (1,639 pregnant and 5,687 postpartum women) were included in the analysis, whereas 721 were excluded due to irrelevant and contradictory responses. These responses were identified based on study-specific definitions, which included one dummy question and two questions regarding disease history and substance use.

### Measurements

#### *Refraining from seeking consultation*

The primary outcome was defined as refraining from seeking consultation at public service facilities (*i.e.,* public service offices, public health centers, and family support centers) or medical facilities. To assess this, we asked the women whether there was a time they wanted to consult about family or parenting issues at public service or medical facilities but could not.

#### *Reasons for refraining from seeking consultation*

We listed 13 predefined reasons for refraining in the questionnaire. Women who reported having refrained from seeking consultation selected all applicable reasons from the list. These reasons included difficulties in taking the child(ren) to the consultation, uncertainty about the seriousness of the problem, not knowing where to seek help, time constraints, skepticism about the problem's solvability through consultation, hesitancy or reluctance, an inconvenient location or unavailability of transportation, problems solved before consultation, feeling blamed by staff, fear of privacy disclosure, past unpleasant experience

with consultation, lack of self-confidence in consultation, and perception of public facility unreliability. We did not provide an open-ended response option for participants to specify reasons other than those listed.

### Women's current concerns

We listed seven types of predefined concerns women had during the survey. They selected all applicable items. The concerns included parenting, financial issues, jobs, physical problems, pregnancy and childbirth, relationships with partners, and family relationships. We did not provide an open-ended response option to specify concerns other than those listed.

### Health literacy

We used the Communicative and Critical Health Literacy scale, which measures interactive and critical health literacy skills (*Ishikawa et al., 2008*). The scale comprised five items, scored from 1 (strongly disagree) to 5 (strongly agree), and we calculated the mean score of these five items for each participant. Subsequently, we categorized the participants into low and high health literacy groups based on the median value.

### Partner support

We asked, "Does your partner (spouse) help you when you are in need?" The women selected one of four response options: all the time, sometimes, not often, and not at all. Subsequently, we categorized the responses "all the time" and "sometimes" as supportive, whereas the responses "not often" and "not at all" as non-supportive.

### Current depression or psychiatric disease

Data were obtained from the question, "Do you have the following disease history?" The women selected one of seven options: (1) never, (2) had it in the past, but it had been cured before pregnancy, (3) got it before pregnancy and currently have it, (4) had it during pregnancy, but it had been cured already, (5) got it during pregnancy, and currently have it, (6) got it after delivery, but it had been cured already, and (7) got it after delivery, and currently have it. Subsequently, we categorized responses (3), (5), and (7) as "currently having the condition" and the responses (1), (2), (4), and (6) as "currently not having the condition."

### Current disability

We asked, "Do you have the following disability?" The women selected one of seven options, which were the same as those in the question about current depression or psychiatric disease. We categorized the responses of having acquired the condition before pregnancy and currently having it, having developed it during pregnancy and currently having it, and having developed it after delivery and currently having it as "currently having the condition." Subsequently, we combined the presence of mental, developmental, visual, hearing, or physical disabilities under the category of current disability.

## Data analysis

A descriptive analysis was conducted to assess the distribution of exposure and outcome variables among the participants.

Multivariate logistic regression analyses were used to identify women's characteristics associated with refraining from seeking consultation at public service or medical facilities. Based on previous studies, we selected the following exposure variables potentially associated with study outcomes (*Grabovschi, Loignon & Fortin, 2013*; *Mi et al., 2022*). These included age groups (18–24, 25–29, 30–34, 35–48), educational attainment (high school or lower, college or higher), household income quartile (Q1; <5 million, Q2; 5–6.7 million, Q3; 6.7–8.5 million, Q4; >8.5 million JPY, [1USD = 110 JPY in July–August 2021], do not know or want to answer), parity (first pregnancy with no child, having one child and currently being in the postpartum period, having one child and currently being pregnant, having two or more children and currently being in the postpartum period, having two or more children and currently being pregnant), health literacy (low, high), partner support (supportive, not supportive), current depression or psychiatric disease (yes, no), and current disabilities (yes, no). Statistical significance was set at $P < 0.05$. Since all participants answered all questions in the survey questionnaire, no missing data were included in our analyses.

For the study outcome regarding whether a woman refrained from seeking consultation at public service or medical facilities, we further assessed the reasons for refraining and calculated the percentage for each of the 13 items. All analyses were performed using Stata BE and SE version 17.

## Ethical considerations

The procedures in this study complied with the ethical guidelines for medical and health research involving human participants enforced by the Ministry of Health, Labor, and Welfare, Japan. Approval for this study was granted by the Bioethics Review Committee of Osaka International Cancer Institute, Japan (20084). Therefore, the study was conducted in accordance with the ethical standards laid down in the 1964 Declaration of Helsinki and its later amendments. Prior to the survey, we obtained informed consent from all participants online. Participants received credit points ("Epoints") as a token of appreciation upon completion of the survey. A research agency anonymized the dataset before sharing it with the researchers, who handled it with strict confidentiality to safeguard participants' privacy.

## RESULTS

Table 1 presents the basic characteristics of the 7,326 women included in the analysis. Among these, 11.9% were experiencing their first pregnancy, 41.9% had one child and were in the postpartum period, 7.2% had one child and were pregnant, and 35.8% had two or more children and were in the postpartum period, and 3.4% had two or more children and were pregnant at the time of the survey. The top three current concerns of women were parenting, financial issues, and jobs, at 60.4%, 54.3%, and 50.5%, respectively. Regarding

study outcomes, 8.6% refrained from seeking consultation at public service facilities, whereas 5.1% refrained from seeking consultation at medical facilities.

Women's characteristics associated with refraining from seeking consultation at public service facilities were those who had one child and were in the postpartum period (aOR 2.07, 95% CI [1.51–2.86]) and those who had two or more children and were in the postpartum period (aOR 1.50, 95% CI [1.08–2.10]) relative to first pregnancy with no child, low health literacy (aOR 1.20, 95% CI [1.01–1.41]), having a non-supportive partner (aOR 1.71, 95% CI [1.34–2.18]), current depression or psychiatric disease (aOR 1.82, 95% CI [1.22–2.72]) and current disability (aOR 2.13, 95% CI [1.19–3.81]) (Table 2). Women's characteristics associated with refraining from seeking consultation at medical facilities included educational attainment of high school or lower (aOR 1.38, 95% CI [1.07–1.78]), those who had one child and were in the postpartum period (aOR 1.57, 95% CI [1.07–2.30]) relative to first pregnancy with no child, low health literacy (aOR 1.42, 95% CI [1.14–1.76]), having a non-supportive partner (aOR 1.71, 95% CI [1.27–2.31]), and current disability (aOR 2.34, 95% CI [1.16–4.71]).

Among women who refrained from seeking consultation at public service facilities ($n = 627$), the top five reasons were "difficulties in taking the child to the consultation" (35.1%), "uncertainty about the seriousness of the problem" (31.7%), "not knowing where to seek help" (28.6%), "time constraints" (27.1%), and "skepticism about the problem's solvability through consultation" (25.7%) (Fig. 1).

Among women who refrained from seeking consultation at medical facilities ($n = 375$), the top two reasons were the same as for public service facilities, "difficulties in taking the child to the consultation" (21.9%), "uncertainty about the seriousness of the problem" (20.5%), followed by "skepticism about the problem's solvability through consultation" (19.5%), "hesitancy or reluctance" (19.2%), and "not knowing where to seek help" (17.3%) (Fig. 2).

## DISCUSSION

In summary, 8.6% had refrained from seeking consultation at public service facilities, and 5.1% had refrained from seeking consultation at medical facilities. Four common factors associated with refraining from both facilities were identified: low health literacy, having a child and being in the postpartum period at the time of the survey, lack of partner support, and current disability. The primary reasons for refraining from seeking consultation were "difficulty in taking the children to the consultation" and "uncertainty about the seriousness of the problem," with a higher percentage at public service facilities than at medical facilities.

Low health literacy was associated with refraining from consulting at public service and medical facilities. Low educational attainment, an established determinant of health literacy (*World Health Organization, 2015*), was also associated with refraining from consulting at medical facilities. In Lavesque's framework of healthcare access, health literacy plays a crucial role in perceiving the need for care (*Levesque, Harris & Russell, 2013*). Health literacy serves as the foundation for individuals to gather, understand, and use information

**Table 1  Basic characteristics of study participants (_n_ = 7,326).**

|  | N | % |
|---|---|---|
| Age |  |  |
| 18–24 | 257 | 3.5 |
| 25–29 | 1,991 | 27.2 |
| 30–34 | 2,942 | 40.2 |
| 35–48 | 2,136 | 29.2 |
| Educational attainment |  |  |
| High school or lower | 1,402 | 19.1 |
| College or higher | 5,924 | 80.9 |
| Household income (by quartile) |  |  |
| Q1 | 1,342 | 18.3 |
| Q2 | 1,729 | 23.6 |
| Q3 | 1,440 | 19.7 |
| Q4 | 1,690 | 23.1 |
| Don't know or want to answer | 1,125 | 15.4 |
| Parity |  |  |
| First pregnancy | 868 | 11.9 |
| One child and currently postpartum | 3,067 | 41.9 |
| One child and currently pregnant | 524 | 7.2 |
| ≥2 children and currently postpartum | 2,620 | 35.8 |
| ≥2 children and currently pregnant | 247 | 3.4 |
| Health literacy |  |  |
| Low | 3,781 | 51.6 |
| High | 3,545 | 48.4 |
| Partner support |  |  |
| Supportive | 6,658 | 90.9 |
| Not supportive | 668 | 9.1 |
| Current depression or psychiatric disease |  |  |
| Yes | 195 | 2.7 |
| No | 7,131 | 97.3 |
| Current disability |  |  |
| Yes | 82 | 1.1 |
| No | 7,244 | 98.9 |
| Types of concern the participating women had at the time of survey (multiple choice) |  |  |
| Parenting | 4,428 | 60.4 |
| Financial issues | 3,979 | 54.3 |
| Job | 3,702 | 50.5 |
| Physical problem | 3,081 | 42.1 |
| Pregnancy and childbirth | 2,302 | 31.4 |
| Relationship with partners | 1,965 | 26.8 |
| Family relationship | 1,447 | 19.8 |

**Table 1** (*continued*)

| | N | % |
|---|---|---|
| Have you ever refrained from seeking consultation at public service facility despite perceiving the need? | | |
| Yes | 627 | 8.6 |
| No | 6,699 | 91.4 |
| Have you ever refrained from seeking consultation at medical facility despite perceiving the need? | | |
| Yes | 375 | 5.1 |
| No | 6,951 | 94.9 |

effectively and access care resources (*Ogasawara & Hashimoto, 2020*). Women with low health literacy may have an insufficient ability to translate their perceived need to seek help. Notably, "uncertainty about the seriousness of the problem," "not knowing where to seek help," or "hesitancy or reluctance" were highly ranked reasons why women refrained from visiting these facilities for consultation, which may indicate low health literacy. Therefore, public service and medical facilities should offer prenatal education on the importance of seeking help, provide information on where and what kinds of issues women can consult about, and create facility environments that women can visit without hesitancy or reluctance.

Women who had at least one child and were in the postpartum period at the time of the survey were less likely to seek consultation at both public service and medical facilities than women in their first pregnancy. The perceived difficulty of bringing their child(ren) to consultation emerged as the primary reason women refrained from seeking consultation at public service and medical facilities, highlighting childcare responsibility as a barrier to accessing services (*Bina, 2020*). Online medical consultation became widely available in Japan in April 2020 (*Ministry of Health Labour and Welfare, 2020*), which improved access to medical services. Conversely, public service facilities are fewer in number than medical facilities, and their consultation services are generally available only on weekdays. Online consultation services provided by public health facilities may help more women seek consultation when they perceive the need. Furthermore, among women with at least one child, those with one child were more likely to refrain from consultation than those with multiple children. This result may represent a difference in parenting experience and knowledge of maternal or child health problems between women with one and those with more children. Similar results have been reported previously (*Brembilla et al., 2018*).

A lack of partner support was also associated with refraining from consulting at public service or medical facilities. Levesque's framework for access to healthcare highlights that social support plays a critical role in seeking and reaching healthcare (*Levesque, Harris & Russell, 2013*). Male partners are frequently identified as the primary decision-makers in seeking treatment for women's distress (*Fonseca & Canavarro, 2017*) and are instrumental in encouraging them to seek help (*Fonseca & Canavarro, 2017*; *Rouhi et al., 2019*). Consequently, women who lack support from their partners may have difficulty accessing public health and medical services. Particularly for women who refrained from consultation due to the challenges of taking their children together, partner support as a caretaker of their children may mitigate this barrier to seeking consultation.

**Table 2  Factors associated with refraining from seeking consultation at public service or medical facility despite perceiving the need.**

| | Public service facility | | Medical facility | |
|---|---|---|---|---|
| | aOR | (95%CI) | aOR | (95%CI) |
| Age | | | | |
| 18–24 | 1.18 | (0.77–1.82) | 1.46 | (0.88–2.41) |
| 25–29 | 1.05 | (0.84–1.31) | 1.29 | (0.97–1.72) |
| 30–34 | 0.94 | (0.76–1.15) | 1.03 | (0.79–1.35) |
| 35–48 | 1.00 | | 1.00 | |
| Educational attainment | | | | |
| High school or lower | 1.13 | (0.92–1.40) | 1.38 | (1.07–1.78) |
| College or higher | 1.00 | | 1.00 | |
| Household income | | | | |
| Q1 | 1.15 | (0.88–1.50) | 1.30 | (0.93–1.82) |
| Q2 | 1.04 | (0.81–1.34) | 1.03 | (0.74–1.43) |
| Q3 | 1.12 | (0.86–1.45) | 1.21 | (0.87–1.70) |
| Q4 | 1.00 | | 1.00 | |
| Don't know or want to answer | 0.99 | (0.74–1.31) | 1.03 | (0.71–1.48) |
| Parity | | | | |
| First Pregnancy | 1.00 | | 1.00 | |
| One child and currently postpartum | 2.07 | (1.51–2.86) | 1.57 | (1.07–2.30) |
| One child and currently pregnant | 1.21 | (0.76–1.92) | 1.19 | (0.69–2.05) |
| ≥2 children and currently postpartum | 1.50 | (1.08–2.10) | 1.30 | (0.87–1.94) |
| ≥2 children and currently pregnant | 0.82 | (0.42–1.62) | 0.41 | (0.14–1.18) |
| Health Literacy | | | | |
| Low | 1.20 | (1.01–1.41) | 1.42 | (1.14–1.76) |
| High | 1.00 | | 1.00 | |
| Partner Support | | | | |
| Supportive | 1.00 | | 1.00 | |
| Not supportive | 1.71 | (1.34–2.18) | 1.71 | (1.27–2.31) |
| Current depression /psychiatric disease disease | | | | |
| Yes | 1.82 | (1.22–2.72) | 1.07 | (0.60–1.91) |
| No | 1.00 | | 1.00 | |
| Current disability | | | | |
| Yes | 2.13 | (1.19–3.81) | 2.34 | (1.16–4.71) |
| No | 1.00 | | 1.00 | |

**Notes.**
aOR, adjusted odds ratio; 95% CI, 95% confidence interval.

Finally, women with disabilities are more likely to refrain from both public service and medical facility visits for consultations. Additionally, women with depression or psychiatric disease were more likely to refrain from seeking consultation at public service facilities. Disabilities and mental health issues are common characteristics of vulnerable populations (*Grabovschi, Loignon & Fortin, 2013*; *Midboe et al., 2020*). The impact of disability on access to consultations may vary by type of disability or mental illness. However, this study did not distinguish between mental disability, developmental disability, visual impairment,

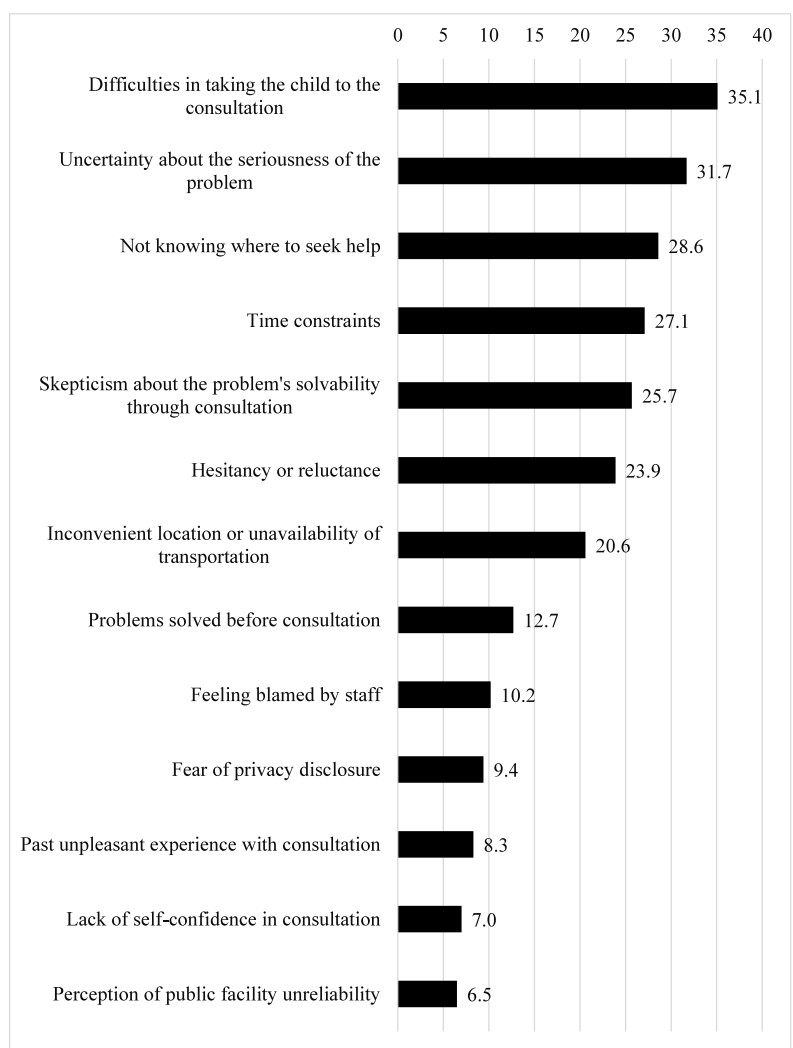

**Figure 1   Reasons that women refrained from seeking consultation at public service facility ($n = 627$).** The display of study results is arranged in descending order.

hearing impairment, and physical disability owing to the small number of women who reported having each disability. Further investigations to assess disability-specific barriers to accessing consultations would be worthwhile.

Our study had some limitations. First, this study used an online survey that involved registered panels pooled by an Internet survey agency. The study sample was limited to those with internet access and registered panelists. Further, 9.0% of participants were excluded from the analysis due to providing irrelevant or contradictory responses. Second, women who were the most vulnerable in accessing consultation might not have participated in this study owing to lack of internet access, low literacy, being occupied with parenting, or poor physical or mental health conditions. For these reasons, the study sample may not represent the general population, affecting the generalizability of the study findings. Third, we did not assess the specific details of family or parenting issues that women

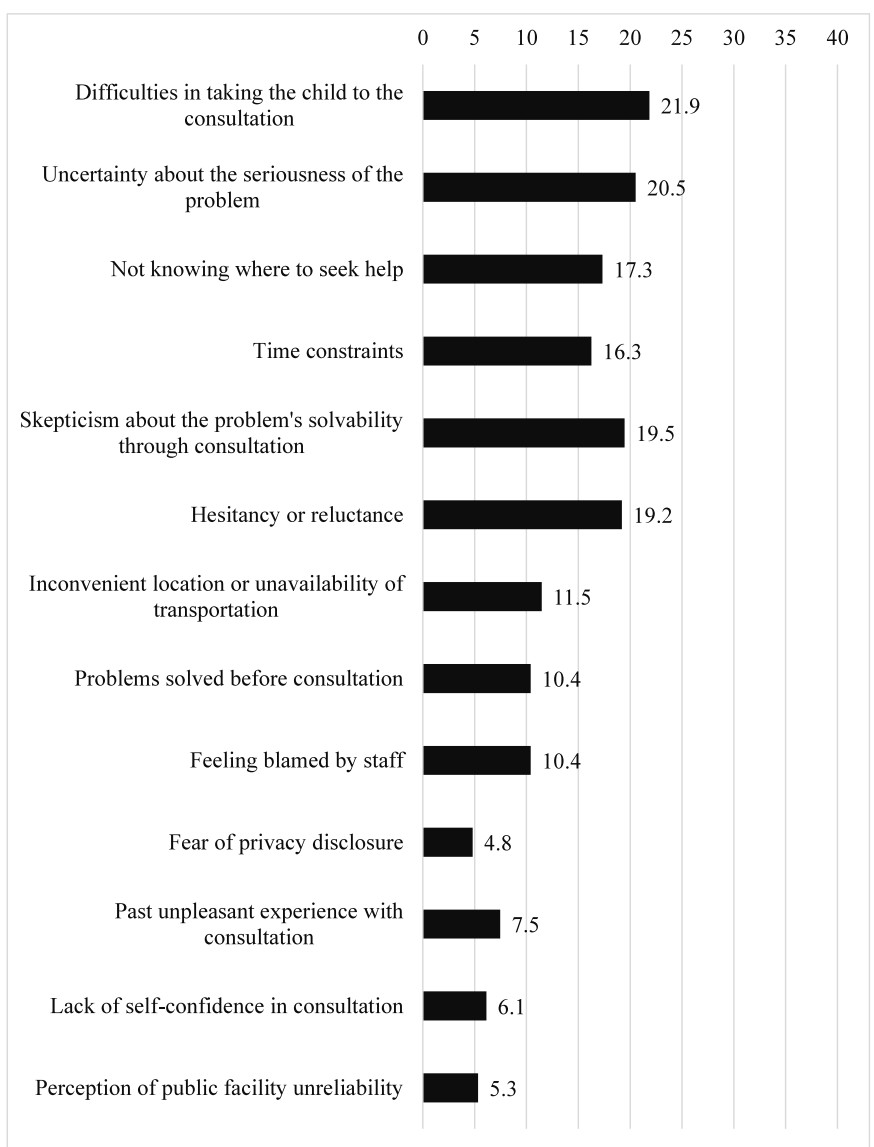

**Figure 2** **Reasons that women refrained from seeking consultation with medical facility ($n = 375$).** The display of study results is arranged in the order corresponding to Fig. 1.

wanted to consult about but refrained from seeking consultation. The absence of this data may limit the interpretation and implications of our findings. Fourth, we might have missed some critical barriers or reasons for refraining from seeking consultation, as we assessed them using a quantitative approach. A qualitative study exploring barriers and reasons not assessed in this study will be worthwhile. Despite these limitations, our study findings provided critical insight into what medical or public service facilities should consider and how they should improve services to help women overcome barriers to seeking consultations.

## CONCLUSIONS

This study revealed that women were less likely to seek support for family and childcare issues, especially under conditions of low health literacy, having a child(ren), lack of partner support, current disability, depression, psychiatric disease, and lower educational attainment. Healthcare workers at public service and medical facilities, central to the community safety net, should provide prenatal education on seeking consultation and pay special attention to these vulnerable women. Furthermore, online consultation systems and facility environments that reduce women's perceived barriers to seeking consultation may encourage their proactive seeking of help.

## ACKNOWLEDGEMENTS

We thank all participants for taking the time to complete our survey while caring for their children. We also thank all members of the JACSIS perinatal survey team. We thank Ms. Tomoka Takano for her valuable comments in improving our work.

### Funding

This work was supported by the Japan Society for the Promotion of Science Kakenhi grants (JP 21H04856), the Japan Science and Technology Agency (JPMJSC21U6), the Intramural Fund of the National Institute for Environmental Studies, Innovative Research Program on Suicide Countermeasures (R3-2-2), the READYFOR Fund for COVID-19 Relief (fifth period, second term 001), the Ministry of Health, Labour, and Welfare (Comprehensive Research on Life-Style Related Diseases including Cardiovascular Diseases and Diabetes Mellitus Grants. 20FA1005) and  grants for the National Center for Global Health and Medicine (24A01). The funders had no role in study design, data collection and analysis, decision to publish, or preparation of the manuscript.

### Grant Disclosures

The following grant information was disclosed by the authors:
Japan Society for the Promotion of Science Kakenhi: 21H04856.
The Japan Science and Technology Agency: JPMJSC21U6.
The Intramural Fund of the National Institute for Environmental Studies.
Innovative Research Program on Suicide Countermeasures: R3-2-2.
READYFOR Fund for COVID-19 Relief.
The Ministry of Health, Labour, and Welfare: 20FA1005.
Grants for the National Center for Global Health and Medicine: 24A01.

### Competing Interests

The authors declare there are no competing interests.
## Author Contributions

- Sumiyo Okawa conceived and designed the experiments, performed the experiments, analyzed the data, prepared figures and/or tables, authored or reviewed drafts of the article, and approved the final draft.
- Kaori Seino conceived and designed the experiments, analyzed the data, prepared figures and/or tables, authored or reviewed drafts of the article, and approved the final draft.
- Hiroyasu Iso conceived and designed the experiments, authored or reviewed drafts of the article, supervision, and approved the final draft.
- Takahiro Tabuchi conceived and designed the experiments, performed the experiments, authored or reviewed drafts of the article, supervision, and approved the final draft.

## Human Ethics

The following information was supplied relating to ethical approvals (i.e., approving body and any reference numbers):

Osaka International Cancer Institute

## Data Availability

The data and the Stata codes we used for statistical analysis are available in the Supplementary Files.

## Supplemental Information

Supplemental information for this article can be found online at http://dx.doi.org/10.7717/peerj.19320#supplemental-information.

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
