# Peer review of "Barriers to seeking consultation at public service and medical facilities among pregnant and postpartum women in Japan: a cross-sectional study"

_PeerJ, doi:10.7717/peerj.19320_

## Round 0.1 · original submission · Minor Revisions

Please revise according to the review comments

·

Basic reporting

The article is well-written with acceptable level of English and well structured. The references are up-yo-date and relevant.

Experimental design

In general the research questions and study objectives are well defined. Sampling was done appropriately and representative of the target population.

Regarding Exposure Variables, how do you measure "depression" among the women? Is it a self-reported data, or there is any way the researchers can access their records?

One of the variables significantly associated with the outcome is "having a non-supportive partner". Please explain how this variable is measured.

Validity of the findings

One of the main finding is "8.6% had refrained from seeking consultation at public service facilities, and 5.1% had refrained 200 from seeking consultation at medical facilities." What is the explanation for the higher reluctance (almost double) of getting help from public services facilities, compared to the medical facilities.

How will the findings be useful in improving healthcare access among women?

Additional comments

Study limitations:
Studying barriers using a quantitative approach limits the findings; they are best explored using a qualitative approach. This needs to be mentioned as one of the study limitations.

Reviewer 2 ·

Basic reporting

Thank you for the opportunity to review this insightful and valuable manuscript. The paper addresses a highly pertinent topic by examining the characteristics of pregnant and postpartum women associated with refraining from seeking consultations in Japan. Congratulations to the authors for addressing such a critical issue. Below, I provide comments and suggestions that may further enhance the strength and clarity of the manuscript.

The introduction is well-structured, concise, and effectively highlights the research gaps in the literature, as well as the significance of addressing this research question. However, given the detailed discussion of depression prevalence across different trimesters of pregnancy and the postpartum period, it may be worthwhile to explore this comparison in subsequent analyses. As a minor detail, in lines 57–58, the authors discuss the number of maternal and newborn deaths but do not specify the mortality rate or provide a timeframe for these figures. Adding this context would enhance clarity and precision.

The Discussion section would benefit from starting with a concise summary of the key findings related to the main study objectives—namely, the prevalence of refraining from seeking consultation and the participant characteristics associated with this behavior. This would provide a clearer context for the subsequent discussion. For instance, the phrase in line 198 (“In summary, approximately 60% of women expressed concerns about parenting”) should not be the opening sentence of the discussion paragraph. Instead, it could be repositioned later in the section to support the interpretation of results.

Experimental design

The variables related to women’s current concerns and reasons for refraining from seeking consultation are introduced in the “Data Analysis” subsection. However, it would be more appropriate to describe how these variables were assessed in the “Variables” subsection. For example, how were concerns evaluated? Was this done using a yes/no question? Similarly, for reasons for refraining from seeking consultation, were these captured through open-ended questions, or was there a predefined list in the questionnaire for participants who answered “yes”?
Regarding the exposure variables, how was “current depression” assessed? Providing details about the measurement or diagnostic criteria would enhance clarity.
In the “Data Analysis” subsection, were current concerns included in the statistical models? It might be valuable to assess whether these concerns are associated with refraining from seeking consultation.
Finally, did all participants answer all the questions in the survey, or was there missing data? Clarifying the extent of missing data and how it was handled (e.g., imputation or exclusion) would strengthen the methodological rigor.

Validity of the findings

While the authors have appropriately mentioned the study's limitations, it is equally important to highlight its strengths.

---

## Round 0.2 · accepted · Accept

Thanks, we are satisfied with your revision.

·

Basic reporting

No comment

Experimental design

Suggested corrections done accordingly

Validity of the findings

Issues clarified.

Additional comments

The authors have made improvement to the latest manuscript and I am pleased to suggest for acceptance.

Reviewer 2 ·

Basic reporting

no comment

Experimental design

no comment

Validity of the findings

no comment

Additional comments

Thank you for sending the revised version of the paper. I have reviewed the changes, and I find them satisfactory. I do not have any further suggestions or revisions at this time.